# Toward Thermochromic VO_2_ Nanoparticles Polymer Films Based Smart Windows Designed for Tropical Climates

**DOI:** 10.3390/polym14194250

**Published:** 2022-10-10

**Authors:** Natalia Murillo-Quirós, Victor Vega-Garita, Antony Carmona-Calvo, Edgar A. Rojas-González, Ricardo Starbird-Perez, Esteban Avendaño-Soto

**Affiliations:** 1Escuela de Física, Instituto Tecnológico de Costa Rica, 30101 Cartago, Costa Rica; 2Centro de Investigación en Ciencia e Ingeniería de Materiales (CICIMA), Universidad de Costa Rica, 11501-2060 San Pedro, Costa Rica; 3Escuela de Ingeniería Eléctrica, Universidad de Costa Rica, 11501-2060 San Pedro, Costa Rica; 4Escuela de Física, Universidad de Costa Rica, 11501-2060 San Pedro, Costa Rica; 5Escuela de Química, Instituto Tecnológico de Costa Rica, 30101 Cartago, Costa Rica

**Keywords:** smart windows, thermochromic, tropics, vanadium dioxide, nanoparticles, polymeric matrix

## Abstract

Thermochromic smart windows have been extensively investigated due to two main benefits: first, the comfort for people in a room through avoiding high temperatures resulting from solar heating while taking advantage of the visible light, and second, the energy efficiency saving offered by using those systems. Vanadium dioxide (VO_2_) is one of the most used materials in the development of thermochromic devices. The countries located in the tropics show little use of these technologies, although studies indicate that due to their characteristics of solar illumination and temperature, they could benefit greatly. To optimize and achieve maximum benefit, it is necessary to design a window that adjusts to tropical conditions and at the same time remains affordable for extensive implementation. VO_2_ nanoparticles embedded in polymeric matrices are an option, but improvements are required by means of studying different particle sizes, dopants and polymeric matrices. The purpose of this review is to analyze what has been regarding toward the fabrication of smart windows based on VO_2_ embedded in polymeric matrices for tropical areas and provide a proposal for what this device must comply with to contribute to these specific climatic needs.

## 1. Introduction

Thermochromic smart coatings have emerged as an alternative regarding energy-efficient buildings [1,2]. Windows are known as the most energy-inefficient components of buildings [3,4], since they allow both the entrance and exit of energy. Thermochromic smart windows can modulate near-infrared radiation, switching from a transmissive state to an opaque state in response to the environmental temperature from low to high [2,5] (see Figure 1), with no extra stimuli required, leading to lower energy consumption.

Vanadium dioxide (VO_2_) is the most widely studied and promising thermochromic material [3,6], being employed as thin film [7,8], nanocomposite, micro-composite and in other structures such as grids or biomimetic designs [9]. VO_2_ shows a reversible first-order phase transition, from a semiconductor monoclinic structure (M) to a metallic rutile-like tetragonal structure (R) when the critical temperature is reached at ca. 68 °C [10,11]. This semiconductor to metal phase change causes a decrease in the electric resistivity and an important increase in the reflectance of the infrared spectrum in the material, with slight changes regarding the visible range of the electromagnetic spectrum [12]. Different crystalline phases have been reported for VO_2_; yet, only the M and R phases show a total reversible semiconductor to metal transformation [13]. It has been found that one way of changing the critical temperature is by doping the VO_2_. The effect on the system varies according to the doping element; having the possibility to vary the transition temperature allows as a consequence to manipulate the temperature in which the system avoids the transmittance of the infrared spectrum [14]. Tungsten (W) is the element that tailors the best system’s transition temperature to ambient values, it reduces the transition temperature by the rate by 20–26 °C per wt. % [15].

VO_2_ coating for smart windows is achieved by sputtering, vapor deposition, ion implantation and similar processes based on vanadium solutions. A disadvantage of these methods is their need for high temperatures (around 500 °C) and rigorous processing conditions; those limitations complicate their scalability. A solution that requires much lower temperatures and less controlled conditions is the synthesis of powdered VO_2_ (M) and its further dispersion into a polymeric matrix and then producing films by coating a substrate [16].

Hydrothermal synthesis is a common pathway for producing metallic oxides. It basically consists of generating a chemical reaction on an aqueous medium to precipitate VO_2_ nanoparticles [17,18,19]. The production of nanoparticles by the hydrothermal method is particularly convenient over other synthesis routes because its final powdered appearance, as will be discussed later, may be added to polymeric films.

To apply it to a surface, such as glass, the powder must be integrated into a polymeric matrix, maintaining the thermochromic properties of VO_2_. The choice of polymer is fundamental to the design of an intelligent window because, in addition to encapsulating the VO_2_ particles, it must have remarkable adherence to the substrate, transparency to visible light and must keep these properties during the lifetime of the window. Thus, the polymer choices made by different groups and their respective considerations are reviewed.

Many of the already mentioned characteristics of VO_2_ have demonstrated its potential in the design of smarts windows; namely, among thermochromic materials, its transition temperature, 68 °C, is the closest to room temperature, and it can easily be changed by doping, so it is possible to design a tunable selective modulator of infrared radiation. In addition, the phase transition of VO_2_ is reversible and its manufacture is simple and inexpensive [14]. Still, some issues need to be resolved to offer it as a commercial solution.

Chang et al. point out the challenges in production of smart windows based on VO_2_, such as an unfavorable brownish-yellow color of the VO_2_ film, the still desirable improvement in optical properties to obtain better light and solar transmittance, the ability to guarantee the stability of thermochromic VO_2_ over time and deciding which approach is the best route for industrial production, given that, currently, the manufacturing methods are maintained at a laboratory scale [2]. As an inorganic thermochromic material, VO_2_ is toxic; therefore, in order to use it, it must be covered, so that people or pets do not touch it. The integration of composites to coat (core–shell) VO_2_ and form nanostructures, the design of multilayer films and surface treatments are presented as possible solutions to explore, not only to overcome the indicated obstacles but also to add functions to the windows, such as self-cleaning or hydrophilicity [20].

Tropical countries have warm temperatures during the whole year, and those are rising due to climate change [21,22]. The data on thermochromic windows for hot climates showed that smart glazing can be effective in tropical regions even as skylights or sunroofs [23]. Moreover, it has been stated that in warmer areas, a lower transition temperature (around room temperature) leads to better energy savings than in places with predominantly cold weather [12,24,25]. Even so, the design of a thermochromic window specifically adjusted to the needs of the tropics has not been currently reported.

The aim of this work is to provide a systematic review of the progress made in the design of thermochromic smart windows based on VO_2_ designed for the tropics. This analysis is covered in Section 1, an introduction to smart windows based on VO_2_ and the tropical conditions that make them ideal for these devices. Section 2 describes in detail the location and climate of the tropics and reviews the research conducted to control the light and temperature conditions inside the buildings constructed in these areas. Section 3 describes the literature associated with the design of VO_2_-based smart windows, from the choice of the synthesis process to the integration of VO_2_ in polymeric matrices for glazing application. Section 4 proposes an ideal device, which, based on the research conducted so far, will meet the requirements of an enclosure built in a tropical climate. Finally, Section 5 presents the main conclusions reached throughout this review. This work summarizes the inherent thermochromic properties of VO_2_ in the selective modulation of infrared radiation transmission without changing the visible electromagnetic spectrum, specifically to design a smart window for tropical weather conditions. This device should respond gradually to environmental conditions through manipulation of the VO_2_ transition temperature using different doping percentages, so that the inhabitants of tropical structures benefit from the comfort and energy efficiency that these devices may provide. 

## 2. Research on Glazing in the Tropics

### 2.1. Geographical and Climatic Situation

The tropical zone extends from latitudes of 10 to 23° (see Figure 2). It is characterized by high rainy rates throughout the year. Its geographical position provides a large amount of solar radiation (nearly 12 h of light throughout the year), which results in high temperature and high level of relative humidity. Tropical regions differ markedly from ones extending from about 35 to 60° latitude, including most of Europe and North America [26].

The United Nations is far from achieving its Sustainable Development Goal to “ensure access to affordable, reliable, sustainable and modern energy for all” by 2030 [27], especially in undeveloped countries. Low energy consumption technologies are key to taking advantage of the natural resources in tropical regions.

Costa Rica is widely recognized as a renewable-energy-based country [28], with a potential need for advanced technological products in the solar area [29]. It is located 9.6° north of the equator, this geographical position means that the country has warm temperatures throughout the year. According to the “Effects of Climate Change on Agriculture” study by ECLAC, the average annual temperature in Costa Rica had a notable increase in the mid-1990s, and since then, although it has dropped, it has not returned to the historical levels prior to this rise. The meteorological models carried out by researchers foresee scenarios of temperature increases for the remainder of this century [30].

Therefore, when thinking about the well-being of the users of tropical houses, there are two major concerns—cooling the space and visual comfort, which means taking advantage of the many hours of natural light that tropic geographical position provides. Most studies of energy efficiency for coatings, both in terms of the theoretical and experimental modeling, pertain to the northern hemisphere with a latitude of 23° [31], which delimits the tropical zone and beyond. However, there is a consensus that thermochromic windows achieve better performance in hot and warm climates than in cold ones [24,31]. Therefore, it is worth delving into the research on windows designed for the tropical climate.

### 2.2. Cooling Spaces Using Material Properties

Analyzing the publications that offered solutions to regulate the temperature using and even improving the conditions of natural lighting in a room, some authors have explored solutions that do not necessarily apply smart windows. For example, Edmonds and Greenup described windows that can be opened, providing natural ventilation, and integrating devices such as light guiding shades. They are basically shelves with opaque and translucent areas that are placed on the outside of a window with different inclinations to cover and redirect the solar lighting to the interior of the room. The researchers also mention the use of angle-selective glazing for radiant heat control [26]. It is worth mentioning that these resources can be useful in subtropical zones (ranging from 23° to 35° degrees of latitude, north or south), since they take advantage of the sun’s inclination with respect to the zenith, but at latitudes close to the equator, the sun is very vertical all year round, so these devices are not very useful.

Nguanso et al. discuss the trends in the construction industry in Thailand. The built-in approach to energy efficiency design does not reflect the current market practice. Their research examines which components work best together to save energy and reduce environmental impact in buildings in the tropical region. They propose a reduction in internal temperatures due to the solar gain by insulating the walls, ceilings and floors. The openings associated with windows will only be double glazed [32].

The approach of Al-Obadai et al. to reducing the heat in houses in southeast Asia has focused on avoiding the passage of solar radiation through the roof, to which they attribute 70% of the heating of a room [33]. This research group explores the passive cooling techniques, such as reflective roof strategy (to slow down the heat transfer into a building) and radiative roof (to remove unwanted heat from a building). Both consist of choosing materials with physical properties that diminish as much as possible the thermal absorptance [33]. Correspondingly, Akbari et al. showed that replacing the conventional surface colors with light colors significantly reduces the infrared radiation and heat absorption [34].

According to Qahtan et al., the success of glazing in the tropics lies in the concept by it transmits the visible light and reflects outwards the short-wave infrared radiation [35]. They point out that spectrally selective glazing might be an appropriate choice for the tropics but is not popular due to its high production cost. As an option, they design a solar control by a sustainable glazed water system, it cools a room by flowing a thin water film over the outer surface of the windows of the building during the day. They achieve two important aims to lower the temperatures with this system: the flow of the water film takes away the heat and lowers the surface temperature, and the water can absorb some solar energy, limiting the passage of thermal energy. They showed p interesting results, showing that water reduces the solar heat (infrared) transmittance and maximizes daylight (visible light) transmittance [35].

### 2.3. Using Smart Materials

Mahdavinejad et al. affirmed that the term smart window has been applied “to any system that purports to have an interactive or switchable surface and has functions like control of optical transmittance, thermal transmittance and absorption and view through the window” [23]. Therefore, by this point, solutions have been proposed that take advantage of material properties, but they cannot be classified as smart devices. In this reference, the group classify the advantages and disadvantages of different kinds of smart windows in a tropical region, e.g., photochromic, thermochromic, thermotropic, electrochromic, liquid crystal device and suspended particle device windows [23]. Regarding thermochromic glazing, the authors claimed that because it responds to heat by partially changing from transparent to opaque the improvement in thermal comfort comes at the expense of the view through the window [23].

Hoffmann and Waaijenberg mentioned that color change could be desirable for tropical greenhouses where there is no problems of plants freezing or suffering from lack of light. On the contrary, the cooling, shading and diffusing effects are key to decreasing the temperature, avoiding direct burning of plants and improving the uniformity of the light intensity inside a greenhouse. Thus, in their research, thermochromic coatings made of liquid crystal acted as shading materials, without exploring the use of vanadium oxides to regulate the temperature [36].

In an investigation carried out in India by Singh et al., the authors studied a large-area smart window fabricated with commercially available thermochromic pigments and gels in combination with invisible mesh electrodes. Their goal was to design a reasonably priced smart window with tunable light and temperature to control the glare and infrared radiation entrance [37]. The device was experimentally tested and showed a reversible change from completely opaque to transparent, near room temperature. It could be activated by the user (using electricity) or by heat from the sun [37]. The glazing developed by this group has an acceptable transparent state below the transition temperature, but when it is reached, the glazing turns totally opaque to visible light; therefore, the people inside the building lose the possibility to see through the window and be illuminated by natural light.

Thermochromic pigments were explored by Hu and Yu. In their case, an adaptive building roof was designed by integrating thermochromic coating and phase change material (PCM) layer, which intelligently control both the solar energy absorption/reflection and thermal energy transfer in the building [37]. The thermochromic coating and PCM used are commercially available. Through simulation, they showed energy savings compared to a traditional roof [37]. In the same way as in the investigation of Singh et al., the thermochromic pigment used turns from a transparent state below the transition temperature to a completely opaque state above it [38]. Thus, their proposal would not be desirable for a window.

A tunable emissivity multilayer thermochromic smart window was designed by Wang et al. [39]. They built a system with a thin film indium tin oxide overcoating deposited on the glass. They added an inner and outer polyethylene layer, and a hydroxypropyl cellulose hydrogel was added to the system. Their results showed that the system provided cooling power to the buildings in the hot season and the developed smart window has a switchable front side and solar emissivity for cold climates too. The last consideration is unnecessary in tropical zones.

Zhou et al. pointed out that “smart window research usually assumes that the sunlight radiates in one direction throughout the year while most regions receive solar radiation at various angles” in different seasons and places [40]. This group constructed a vanadium-dioxide-based thermochromic smart window, considering the solar elevation angle. Three-dimensional printing technology was employed to fabricate the tilted microstructures for modulating solar transmission dynamically. They claimed that the tilt, thickness, spacing, width and tungsten doping percentage in VO_2_ can be customized according to the average temperature and solar elevation angle variation of the sun in a determined latitude. Through energy consumption simulations in different cities near the Tropic of Cancer, they showed good results for this device, which they proposed may open a new era of real-world-scenario smart window exploration.

According to Shen et al., triggering the metal to isolator transition (MIT) by Joule heating upon voltages applied on VO_2_-based devices avoids the degradation of thermochromic performance and the significant reduction in MIT temperature to room temperature [41]. Using this system, they achieved a synergetic result of tungsten doping and hysteresis behavior of MIT in VO_2_. The optimal infrared blocking performance could be retained at a reduced cost of energy consumption and at an ambient temperature down to 47 °C, which coincides with the glass window temperature in the summer in subtropical and tropical regions. The authors concluded that an optimal infrared blocking performance of VO_2_ films needed low and even zero energy consumption [41].

Al-Obadai et al. and Hoffmann and Waaijenberg established that, in practice, a restriction on using thermochromic devices is their high cost [33,36]. This is one of the challenges to be overcome in the design of window coatings.

From the architectural point of view, Heidari et al. mentioned that it is not real that applying a single device will be the solution for the control of energy in developing countries. Energy efficiency may be achieved using a combination of several systems and long-term strategies [42].

## 3. VO_2_ in Thermochromic Films

### 3.1. Advantages of Dispersed VO_2_ Nanoparticles over a Continuous Thin Film

In the following section, important figures of merit of thermochromic coatings are introduced. We briefly discuss the advantage of using vanadium dioxide (VO_2_) nanoparticles embedded in a dielectric matrix (e.g., a polymeric matrix) instead of continuous VO_2_ thin films for improving the performance of thermochromic smart windows. A more detailed description can be found elsewhere [43,44].

Here, the two relevant aspects to consider are the amount of solar and luminous radiation that can be transmitted by the device, which could be described by the integral luminous $T_{\mathrm{lum}}$ and solar $T_{\mathrm{sol}}$ transmittances, respectively. $T_{\mathrm{lum}}$ and $T_{\mathrm{sol}}$ are given by
(1)$$T_{\mathrm{lum},\mathrm{sol}}=\frac{\int \varphi_{\mathrm{lum},\mathrm{sol}}\left(\lambda \right)T\left(\lambda \right)\;d\lambda }{\int \varphi_{\mathrm{lum},\mathrm{sol}}\left(\lambda \right)\;d\lambda },$$
where $\varphi_{\mathrm{lum}}\left(\lambda \right)$ is the relative luminous efficiency of the eye [45] (see Figure 3a), and $\varphi_{\mathrm{sol}}\left(\lambda \right)$ is the solar irradiance at sea level, which is typically depicted by the AM 1.5 standard spectrum [46] (see Figure 3a). In general, a transparent glazing is desired, which corresponds to a high $T_{\mathrm{lum}}$ value. Regarding $T_{\mathrm{sol}}$, the important quantity in thermochromic coatings is the contrast $\Delta T_{\mathrm{sol}}$ between the integral luminous transmittance in the semiconducting $T_{\mathrm{sol}}\left(\tau <\tau_{C}\right)$ and metallic $T_{\mathrm{sol}}\left(\tau >\tau_{C}\right)$ state, which is defined as follows [44].
(2)$$\Delta T_{\mathrm{sol}}=T_{\mathrm{sol}}\left(\tau <\tau_{C}\right)-T_{\mathrm{sol}}\left(\tau >\tau_{C}\right),$$
where τ is the temperature at the glazing, and $\tau_{C}$ is the critical temperature. The higher the value of $\Delta T_{\mathrm{sol}}$, the more significant energy-saving effect can be achieved by the thermochromic smart window.

Figure 3b,c depict the typical spectral transmittance and reflectance of a continuous 50 nm thick VO_2_ thin film in the semiconducting and metallic state [47]. On the other hand, Figure 3d,e show the calculations of the same quantities in the case of VO_2_ nanospheres dispersed in a 5 µm thick dielectric matrix with a volume factor of 0.01 and an effective thickness of 50 nm [48].

The reflectance in the nanoparticle case does not show a significant variation between the metallic and semiconducting state (see Figure 3e), in contrast with what is observed for the thin film (see Figure 3c). Thus, the transmittance changes in Figure 3d can be ascribed mainly to absorption.

As depicted in Figure 3a, the relative luminous efficiency of the eye is only relevant within a spectral region between about 400 and 700 nm. One of the main drawbacks of the continuous thin film case is the low $T_{\mathrm{lum}}$ due to the low transmittance (of about 40 to 50%) between about 400 and 700 nm. By comparison, at the same spectral region, the dispersed nanospheres case presents higher transmittance (see Figure 3d), which yields a higher $T_{\mathrm{lum}}$ value with respect to that of the continuous thin film.

For the continuous thin film, the difference in transmittance between the semiconducting and metallic state is more pronounced at around 2500 nm (see Figure 3b), which corresponds to a spectral region with low solar irradiance (see Figure 3a). Regarding the nanospheres case, an interesting aspect to notice in Figure 3d is the minimum of transmittance in the metallic state between about 1000 and 1500 nm. An important consequence of this feature, which can be assigned to the plasmonic absorption effects [49], is an increase in $\Delta T_{\mathrm{sol}}$ with respect to that of the continuous thin film. This is because for the nanospheres case, the region of higher difference between the transmittances of the semiconducting and metallic states shifts toward lower wavelengths in comparison to the continuous thin film case, which is a spectral region with higher solar irradiance than that at about 2500 nm (see Figure 3a).

In summary, for practical applications, it is preferable to use a thermochromic glazing based on a dielectric matrix with dispersed VO_2_ nanoparticles rather than the continuous thin film version. For comparable configurations in particular, the former gives higher $T_{\mathrm{lum}}$ and $\Delta T_{\mathrm{sol}}$ values than the latter.

### 3.2. Hydrothermal Synthesis of Monoclinic VO_2_ for Thermochromic Applications

Hydrothermal synthesis consists of generating a chemical reaction in an aqueous medium, and it is a method of producing metal oxide particles. It was developed in the early 1970s by researchers such as Matijevic, who used an autoclave at room temperature and atmospheric pressure to synthesize chromium hydroxides [50]. From his work, it seems that the first step of the reaction is the hydrolysis of the metal salt solution to produce hydrated oxide particles, and the second part is its dehydration to produce the metal oxide. However, the hydrated metal oxides are also produced, and the reaction rate is low because the temperature is usually not high enough for dehydration. It is advisable to carry out hydrothermal processes at the highest possible temperature and pressure [51].

The synthesis of nanoparticles for thermochromic oxides has been reported using the hydrothermal method through different routes [52,53], one of its advantages is that it allows doping the material in the same production of vanadium oxide [54]. On the other hand, due to the multiple oxidation states of vanadium and its many polymorphic forms, it could be quite a challenge to prepare one single phase of vanadium oxide [55]. Variables such as temperature, time, pressure, precursor addition rate, reducing agents and others should be carefully controlled [53,56,57] (see Table 1). Even though the product of hydrothermal synthesis is usually a mixture of two or more of these forms, it is possible to bring them to VO_2_(M) by means of an annealing process [58]. This implies a second step to achieve the synthesis of the thermochromic material [4]. One-step hydrothermal synthesis of monoclinic VO_2_(M) powders has become a research focus to avoid the aggregation and growth of nanoparticles during a second step heat treatment [18].

As can be seen in Table 1, the literature shows that vanadium oxides can be synthesized by hydrothermal methods using several processes, such as reduction in pentavalent vanadium compounds (V_2_O_5_ or NH_4_VO_3_) with different reducing agents, as well as homogeneous precipitation of VOSO_4_ under mild hydrothermal conditions [53]. The hydrothermal method usually requires temperatures around 240 °C [18] and reaction times of a few hours up to two days [18,19,59,64] to obtain VO_2_(M) in a one-step hydrothermal reaction process [18]. The control of pH and inert atmosphere are key factors to achieving the synthesis of VO_2_(M), since the acid conditions and oxygen lead to non-thermochromic forms of vanadium oxides. Some investigations report pH values from neutral to slightly basic [64], indicating that the gradual and uniform rise in pH can result in the nucleation and growth of uniformly nanosized particles [19].

A one-step direct synthesis of pure monoclinic VO_2_ nanoparticles by continuous hydrothermal flow synthesis has been reported, which is a variation of the hydrothermal method described above. It used a jet of supercritical water mixed with dissolved metal salts of the precursor at room temperature. It allows the reproduction and scalability of the system for manufacturing purposes [65].

### 3.3. Methods Used to Include VO_2_ NPs in Polymeric Films

VO_2_ nanoparticles can be used as thermochromic materials because of the changes in optical properties as a reaction to the temperature. This results in windows that selectively modulate the light that passes through them. However, constructing a window that includes VO_2_ nanoparticles (NPs) as part of it is a challenge to be studied.

In this section, we classify and discuss the different methods used to include VO_2_ NPs in polymeric matrices to finally form a composite material that could work as a thermochromic window. Table 2 summarizes the relevant references classifying the methods of obtaining the VO_2_ nanoparticles, the type of polymer, film thickness and nanoparticles size.

Since 2013, efforts have been made to produce VO_2_ glazing on a surface of 2.72 m^2^ [66]. A PET film was covered with VO_2_ and finally adhered to a glass substrate, creating a composite material with a phase transition temperature of 41.3 °C. However, there is limited information on the film fabrication methodology [9]. In this reference, polyvinylphenol was also used as a polymeric matrix together with commercial VO_2_ particles that were treated with an ultrasonic device to decrease its size and perform IR modulation.

VO_2_ NPs—with a size between 10 and 200 nm—have been combined with elastomeric polymers, such as polydimethylsiloxane (PDMS), to explore the influence of film thickness on solar energy modulation [68], as the film could be stretched and released to control the width of the path that solar irradiance must go through. Another polymer widely used on thermochromic windows is polyvinyl butyral (PVB), as it is commercially available and normally utilized for these kinds of applications [82]. The same polymer, PVB, functions as a matrix in which the VO_2_ particles are placed; the NPs are dispersed using an ultrasound device, and the suspension is stirred to be deposited later onto a glass substrate via spin coating, followed by a thermal treatment [69].

Polyvinylpyrrolidone (PVP) has been employed to form a matrix in which VO_2_ nanoparticles were embedded [72]. Once the NPs were synthesized, they were mixed with PVP and some solvents to form a castable slurry, which was later applied to a flexible substrate (PET) through a roll-to-roll method. Then, after applying heat, the solvents evaporated and formed a solid film with well-dispersed VO_2_ nanoparticles. Moreover, PVP has been used as a film promoter during a sol–gel synthesis process with annealing, in which an ambient rich in NH_3_ was used to prevent the further oxidation of VO_2_ [70].

VO_2_ nanoparticles must be stabilized because of their tendency to oxidize relatively fast [73]. According to this reference, one possibility of avoiding the VO_2_ → V_2_O_5_ chemical reaction is to cover the particles with polyethylene (PE) shell, which results in changes not only to the structure but also to the optical properties. Additionally, the covered VO_2_ NPs are combined with ethylene vinyl acetate (EVA), which functions as a matrix, forming films of about 0.3 mm.

PMMA has also been mixed with VO_2_ NPs, being used as a polymeric matrix via electrospinning and hot pressing to develop nanocomposite films [71]. These films can be stacked to improve mechanical behavior. Another approach was followed in Ref. [77]. After being synthesized with a hydrothermal method, the VO_2_ NPs were dispersed in PMMA using an ultrasound with the objective of providing high lifetimes for the VO_2_ NPs. According to the authors, the PMMA final structure was highly crosslinked after being irradiated with UV radiation (220–380 nm). The PMMA/VO_2_ film was deposited on a Polyethylene terephthalate (PET) layer via the blade-coating method. With the idea of creating a thermochromic window with an antimicrobial functionality, PMMA has been co-polymerized via UV curing with N,N-dimethyl-N-{2-[(2-methylprop-2-enoyl)oxy]ethyl}undecane-1-aminium bromide (dMEMUABr), where the VO_2_ NPs were dispersed. Using an adhesive–coated PET, the previously synthesized VO_2_ NPs on a mica substrate were removed and placed on the flexible PET layer, forming the PET/VO_2_/mica material [79]. The solar modulation of this system reached 36.1% at 25 °C, and 90.3% of the bacteria analyzed were killed.

With the objective of developing a smart energy-efficient window that included privacy as another functionality, research inspired by cephalopod skin has been developed [76]. A polydimethylsiloxane (PDMS) elastomeric matrix with dispersed VO_2_ nanoparticles was prepared. Additionally, a polyvinyl alcohol (PVA) layer was added on top by drop casting. Here, it is important to point out that the PDMS composite material is stretched up to 175% to recreate the structure of the film to provide privacy. Moreover, PVA and PDMS were chosen due to their relatively high transparency on the UV–NIR spectrum.

To achieve an infrared stealth material, antimony tin oxide was combined with VO_2_ and polyacrylonitrile, as described in Ref. [80]. Using an electrospinning and sintering process, the composite nanofibrous material was able to decrease its IR emissivity around 68 °C, corresponding to the phase change from a monocyclic unit cell to a rutile phase of the VO_2_ [81].

Even though several formulations have been reviewed, VO_2_ particle concentration and the effect of the polymer matrix and formulation on the thermochromic performance must be addressed to optimize the window properties for tropical weather.

### 3.4. Why Is Necessary to Reduce the Particle Size?

VO_2_ particle size control is critical for window applications where transparency is a requirement. To achieve an ideal control regarding the modulation of infrared radiation with minimal effect on visible light transmittance, the literature indicates that the particles must be less than 100 nm in size [16]. However, it was reported that even sizes of about 50 nm may be reached [65]. It has been reported that the size of the VO_2_ NPs can be controlled by using different annealing temperatures, making it possible to change the optical properties of the deposited films, as well as enabling large-scale synthesis [83].

The size of the particles resulting from the synthesis is on a scale of microns, so it can be detected by the human eye and also produces visible light scattering, which results in a decrease in visual transparency through the glass [18,84].

From the optics point of view, there are difficulties in maintaining the high transmittance of transparent polymeric matrices when the particles of inorganic materials are dispersed in them due to the difference in the refractive index of the particles and the polymeric matrix. Another factor on which transparency strongly depends is the radius of the scattered particles. It is possible to minimize the scattering losses in systems containing fine particles of one to two orders of magnitude smaller than the wavelength of light [85].

In reducing the particle size, a balance must be reached, since reducing it to very small sizes enhances the formation of agglomerates, which deteriorates dispersion in the polymeric matrix [85].

Computational modeling indicates that the loss of light intensity due to particle scattering in the polymeric matrix can become negligible if the particle size is reduced below 100 nm [16], which is consistent with the information shown in Table 2.

#### VO_2_ Particle Size Reduction Methodologies

In general, when choosing a technique for particle size reduction in solid materials, the principal aim is achieving a narrow particle distribution, maintaining repeatability along the process. Mechanical methods seem to fulfill this requirement. As an example, in the pharmaceutical industry, those techniques are used to improve the solubility of drugs. This strategy results in increased surface area, increased saturation solubility and decreased diffusional distance, all of which lead to an increase in the extent and the rate of dissolution [86]. Additionally, for the mining industry, particle size reduction is necessary to obtain a final product with fewer pollution agents. The ideal target particle size for comminution is the liberation size, the size about which the valuable mineral can be effectively separated from the gangue by physical or chemical methods [87].

Wet jet milling was developed as a novel mixing and dispersion method for a suspension, in which the agglomerates are pulverized by turbulent and shear flows generated from the high-speed injection into an exclusive canal [88,89]. It has been reported as a tool for accomplishing a significant particle size reduction in solids [89].

A wet jet variation is called jet milling. In this micronization method, high-velocity compressed air streams are injected into a chamber where the originally raw materials are fed with a rate-controlled feeder. As the particles enter the air stream, they are accelerated and caused to collide with each other and the wall of the milling chamber with high velocities [90]. The pressure regulation in the chamber allows to introduce the energy into the system. As a general rule, the higher the pressure, the higher the particle velocity, which leads to a broad particle size distribution by the end of the process. In Ref. [91], different tests with tungsten powder, with an average size of 1µm, 3 µm, 5 µm, 10 µm, were performed. The results showed that particle size reduction was achieved for most of the sample, and above all, the particle size distribution became narrower, which is the goal when performing the jet milling technique.

Another approach for particle size reduction is the combination of two or more methods because each of them has its own advantages. For example, in Ref. [92], an aluminum–iron blend was used to create nanoparticles with jet milling and ball milling techniques. The greatest reduction reached approximately 50 nm, in contrast to irregularly shaped and sized particles in the micron range achieved using ball milling alone. 

As mentioned before, ball milling helps in particle size reduction. In this device, a suitable powder charge (typically, a blend of elemental) is placed in a high-energy mill, along with a suitable milling medium [93]. The goal of milling is to reduce the particle size and blending of particles in new phases. The balls may roll down the surface of the chamber in a series of parallel layers, or they may fall freely and impact the powder and balls beneath them [94]. It was shown that these strategies can be applied to a wide group of materials. In Ref. [95], quartz (SiO_2_) from two different locations was treated in a ball-milling process at durations of 4 min, 120 min, 960 min and 1920 min. Afterward, the SEM and laser scattering results showed an effective particle size reduction in each period. For example, 100 µm average particle size was achieved at 4 min versus 10 µm at the longest time [95].

Despite all the techniques described above, ball milling seems a suitable technique to be applied in the fabrication of thermochromic windows, and also, it has a good projection for industrial-scale use. The advantages of this technique include cost effectiveness, reliability, ease of operation, reproducible results due to energy and speed control, applicability in wet and dry conditions on a wide range of materials (e.g., cellulose, chemicals, fibers, polymers, hydroxy-apatite, metal oxides, pigments, catalysts) [96].

## 4. Conceptual Device

The characteristics of a vanadium dioxide VO_2_-based thermochromic smart window designed for tropical areas should focus on the specific need for thermal comfort by modulating the internal temperature of the enclosures, which, in a tropical area, during the day, tend to reach temperatures well above 25 °C. Temperature regulation must be achieved by avoiding affecting the transmittance of the visible light spectrum, since, in the tropics, the daylight lasts approximately 12 hours throughout the year, which makes it possible to avoid dependence on artificial lighting. According to Wang et al., there is research to be conducted on these two fronts, mainly studying the dopants that both reduce VO_2_ transition temperature and improve the optical properties of the coatings [97]. In this way, we could design devices that really contribute to the energy efficiency of a room.

In the case of nanoparticles embedded in a polymer matrix, there are some advantages regarding the flexibility [98], and the tailor design of the optical properties of the final composite film [20]. On the one hand, it is possible to adjust the particle size, the level of doping, and therefore, the transition temperature [64]. The VO_2_ particle concentration and the addition of liquid crystals may enhance other color hues [99]. The use of a VO_2_ particle size distribution or same-sized particles within the polymer matrix with different doping levels may allow adjusting the dynamics of the window to tune its modulation at different temperatures.

Figure 4 shows the scheme of three possible window designs that can be reached. In addition to the requirements indicated in the previous paragraph, needs such as the protection of coating to avoid VO_2_ degradation and protecting the user must be considered because of the toxicity of vanadium compounds. Figure 4a presents a multilayer device in which the window glass is the substrate to which the VO_2_-active material coating would be adhered and sealed with a protective coating that could have anti-reflection, self-cleaning properties, if necessary, depending on the requirements of the customer [97]. It can be tailored to have some specific coloration and even selective dispersion characteristics, taking advantage of the properties of liquid crystals [100].

An alternative approach of laminating two glass panels with the active material in between is shown in Figure 4b. In this configuration, the optics of the window may be affected due to the thick glass. However, this presents various advantages, as it can be improved with self-cleaning and antireflection coatings, but more importantly, it gives the possibility of using colored glass with different hues, which opens vast prospects for architecture facade design and art installations [101] without neglecting the comfort and energy saving.

Figure 4c offers one further step on scaling up the window manufacture. In this case, the protective coating layer would be made of flexible glass or plastic mesh, which would act as a substrate for the active material, so the coating process could easily become a roll out process, which could even be extended to the application of the self-cleaning of antireflective coatings. There are two possibilities for the substrate cover material: a PET plastic web or a flex glass. PET plastic is more affordable but may have the inconvenience of its optical transparency being inferior to the flex glass. Potentially, the roll out design allows easy cover of flat or non-flat windows, so it can be used to retrofit windows already built, which makes this system very versatile in terms of usage and costs and with a minimum weight footprint [67].

There are investigations of methods to prevent the degradation over time of VO_2_ compounds, as they gradually transform into V_2_O_5_, which is thermodynamically more stable but not thermochromic. W_2_O_3_ coatings have been shown to be a barrier to vanadium oxidation at temperatures and humidity consistent with those of the tropics [2], and could be included in the configurations proposed in Figure 4a,c.

It is important to consider that in the economic reality of tropical areas, the proposed solutions must be adaptable to different budgets in order to be accessible to as many people as possible. 

Currently, the field of smart thermochromic windows based on vanadium is intensively studied but still has many possibilities for new contributions. The objective of this review is to summarize the strategies that, integrated with new ideas in the construction of buildings, provide options for quality of life and energy savings for the inhabitants of the tropics.

## 5. Conclusions

In this review, we bring together the most relevant research related to thermochromic smart windows to shed light on the challenges and opportunities in developing these devices for countries with a tropical weather. The importance of using vanadium dioxide (VO_2_) thermochromic nanoparticles dispersed in a polymeric matrix is highlighted as an alternative to increase its stability in practical applications, as well as contributing to improve smart window energy-saving features. The methods reported in the literature for the synthesis of VO_2_, the polymer matrices normally utilized and the deposition procedures are thoroughly described. Additionally, the configuration and the material functionalities proposed for an ideal conceptual device suited to the weather conditions found in the tropics are discussed. Despite all the challenges in the design of thermochromic VO_2_-polymer-based smart windows exposed in this review, there are weighty possibilities for VO_2_-polymer films regarding their potential use for thermal and light comfort in tropical areas, where the sun shines practically vertically and for many hours throughout the year. As a result, this article contributes to the quest for a device that could help bring thermochromic windows in a practical application in the near future for the tropics.

## Figures and Tables

**Figure 1 polymers-14-04250-f001:**
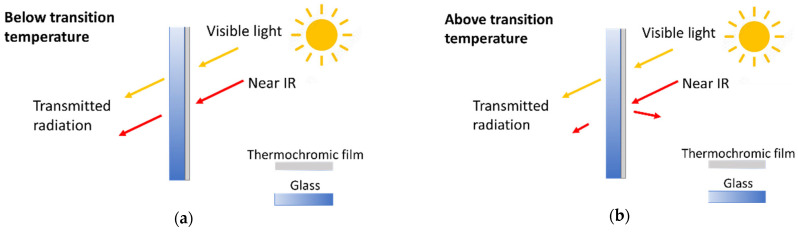
Schematic operation of a smart thermochromic window: (**a**) Below transition temperature, (**b**) Above transition temperature.

**Figure 2 polymers-14-04250-f002:**
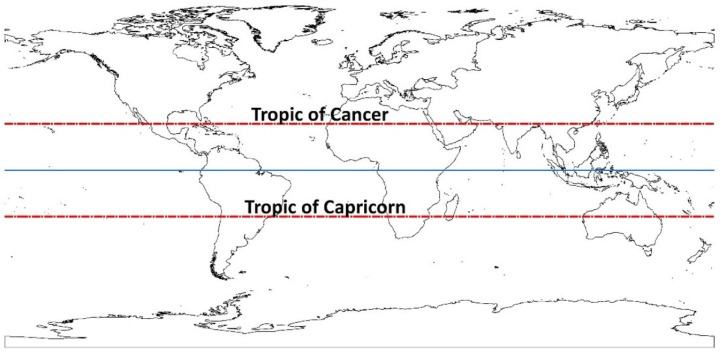
Geographical areas included in the tropics (red lines) and Equator (blue line).

**Figure 3 polymers-14-04250-f003:**
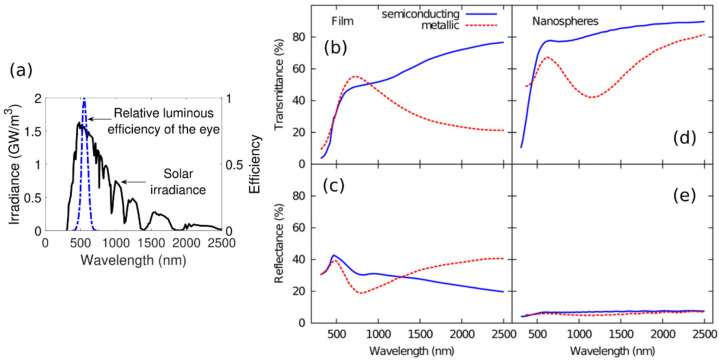
(**a**) Spectra portraying the relative luminous efficiency of the eye [45] and the solar irradiance at sea level for a clear weather with the sun standing 37° above the horizon, corresponding to the AM 1.5 standard spectrum [46]. Spectral transmittance (**b**) and reflectance (**c**) of a VO_2_ thin film (50 nm thick) in its semiconducting and metallic state [47]. Calculation for the spectral transmittance (**d**) and reflectance (**e**) of a 5 µm thick dielectric matrix with a dispersion of VO_2_ nanospheres with an effective thickness of 50 nm for the semiconducting and metallic state [48]. Panels (**b**)–(**e**) were adapted from Ref. [44], Copyright 2016, with permission from Elsevier.

**Figure 4 polymers-14-04250-f004:**
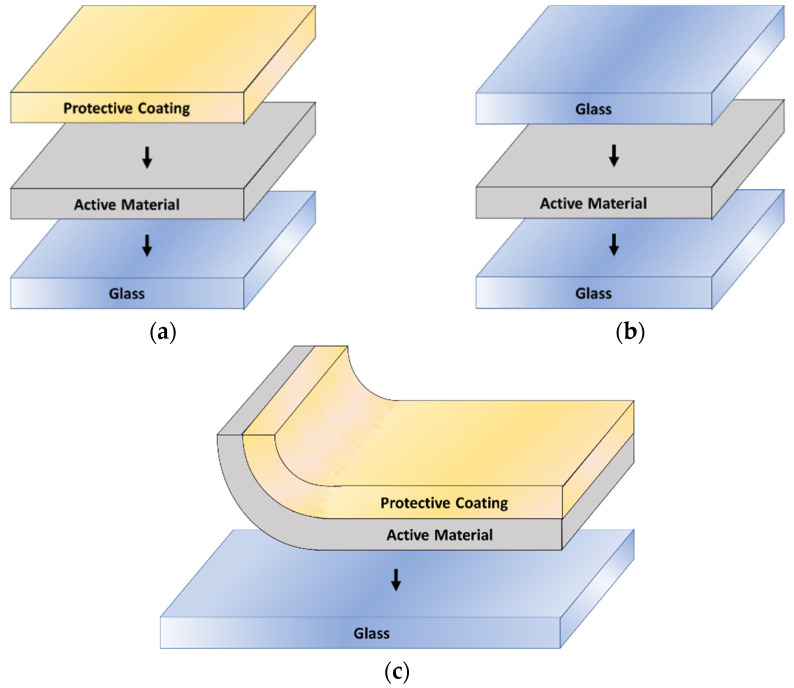
Diagrams of possible useful devices for the design of intelligent thermochromic windows. Manufacture (**a**) monolithic, (**b**) multilayer, (**c**) roll out system.

**Table 1 polymers-14-04250-t001:** Common hydrothermal conditions, crystallography data and comments on VO_2_ polymorphs (from Ref. [59] with permission from John Wiley and Sons).

Common Reaction Conditions	Polymorphs	Unit cell Parameters a, b, c (10^−10^ m)	β (°)	Space Group	Comments	Ref.
V source: V_2_O_5_, VOSO_4_, NH_4_VO_3_	VO_2_ (B)	12.054, 3.693, 6.424	106.96	C2/m	It has a reversible structural switch between the crystalline and amorphous phase under high pressure.	[60]
Reductant: H_2_C_2_O_4_, N_2_H_4_	VO_2_ (A)	8.450, 8.450, 7.678	90	P4_2_/ncm	It has an intermediate phase between VO_2_(B) and VO_2_(R), and a reversible phase transition at ≈ 162 °C.	[15,19]
Surfactant: PVP *, PEG **	VO_2_ (M)	5.752, 4.538, 5.383	122.64	P2_1_/c	The most widely studied inorganic thermochromic material, and most applications are based on the MIT.	[19]
pH regulator: HCl, HNO_3_, H_2_SO_4_, CO(NH_2_)_2_	VO_2_ (R)	4.554, 4.554, 2.856	90	P42/mnm	It has a high temperature phase of VO_2_(M) and a reversible phase transition with VO_2_(M) at τ_c_ ≈ 68 °C.	[61]
Doping element: W, Mo, Mg	VO_2_ (D)	4.597, 5.684, 4.913	89.39	P2/c	A new phase was first reported by Xie et al., and it can be transformed into VO_2_(M) at temperature as low as 300 °C.	[62]
Temperature: ≈180–260 °C Time: From a few hours to a few days	VO_2_ (P)	4.890, 9.390, 2.930	90	Pbnm	It was synthesized using a simple chemical reaction route by Wu et al., and it can be transformed into VO_2_(M) by fast annealing.	[63]

* PVP: polyvinylpyrrolidone, ** polyethylene glycol.

**Table 2 polymers-14-04250-t002:** Relevant references for combinations of polymers and VO_2_ nanoparticles.

Synthesis Method	Polymer Used	Notes	Film Thickness (nm)	NP Size (nm)	Year	Ref.
	Polyethylene terephthalate (PET)				2013	[66]
	Polyvinylphenol		1 mm	950	2014	[9]
hydrothermal and pyrolysis				22	2015	[67]
hydrothermal	Polydimethylsiloxane (PDMS) (matrix)	Films were dried and cured		10–200	2016	[68]
hydrothermal	Polyvinyl butyral (PVB) (matrix)	The film was deposited by spin coating and later dried up.		40	2017	[69]
sol–gel	Polyvinylpyrrolidone (PVP) (film promoter)	PVP was decomposed during annealing.	100	17–20	2017	[70]
hydrothermal	Poly(methyl methacrylate) (PMMA) (matrix)	Electrospinning and hot pressing used for layer formation.		30	2017	[71]
thermal treatment of bead mill	Polyvinylpyrrolidone (PVP) (matrix) and Polyethylene terephthalate (PET) as substrate			30–60	2018	[72]
hydrothermal	Polyethylene (PE) (coating)/EVA (matrix)	PE is used to stabilize VO_2_.	300,000	17.4	2019	[73]
	Poly(methyl methacrylate) (PMMA)	Wood template used as a base.			2019	[74]
hydrothermal	Polyaniline (PANI)	The use of PANI maintains VO_2_ thermochromicity.		-	2019	[75]
	Polyvinyl alcohol and Polydimethylsiloxane (PVA/ PDMS) (matrix)	Polymer chosen due to high transparency on the Vis and IR regions.		60	2020	[76]
hydrothermal	Poly(methyl methacrylate) (PMMA) (matrix)	The film was deposited using blade coating method.	4000	50–80	2020	[77]
hydrothermal	dMEMUABr copolymerized with PMMA			1400 (length) 149 (width)	2020	[78]
annealing	Polyethylene terephthalate (PET) (substrate)	Direct transfer was used for film deposition.	75,000	3.7	2021	[79]
	Polyacrilonitrile (PAN) (matrix)	Electrospinning.			2021	[80]
hydrothermal	Poly(N-isopropyl acrylamide)			20–50	2021	[81]

## Data Availability

Not applicable.

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
