# Peer review of "Toward Thermochromic VO2 Nanoparticles Polymer Films Based Smart Windows Designed for Tropical Climates"

_polymers, 2022, doi:10.3390/polym14194250_

Round 1

Reviewer 1 Report

Review entitled "Towards thermochromic nanoparticles based smart windows 2

 designed for tropical climates" has focused on VO2 based polymer films for controlling temperature using thermocromic effect. It start with introduction, condition of radiation interaction and conceptual design and conclusion. 

Review covers 115 references. Since the whole review article is focused on VO2, its title is not correct. It should clearly mention VO2 polymer films in the title. 

I have found following major points which need to be clarify or included. 

What is the objective of the review article. 

There should be clear objectives to justify the need of this review article as there are 2 review articles on same line 

https://doi.org/10.1039/C7TC05768G

Adv. Manuf. https://doi.org/10.1007/s40436-017-0209-2

These are cited in the manuscript but not at right place. It is also important to include future challenges and advantages of VO2 over other materials.

  Compare the performance of VO2 with other thermocromic film materials. 

Authors should also comment on other importance factors such as stability, economic aspects etc.  

Author Response

Attached is our response.
Best regards.

Reviewer 2 Report

This review summarized thermochromic VO2 smart windows towards tropical climates. On the whole, this review is well organized and contains enough information, and will be very interesting to those who working in this field. This review can be accepted for publication after minor revision.

1) The advantages and disadvantages of different synthesis methods for VO2 nanoparticles are better addressed. 

2) In the aspects of the hydrothermal preparation of VO2, another characteristic work, that is "J Mater Chem A, 2014, 2, 4520-4523" should be added, so that the readers could have a better and overall understanding of the large scale synthesis of VO2 nanoparticles.

3) As is known, the metal-insulator transition of pure VO2 is at 68 °C, which also limits the practical applications of smart window that were used at room temperature. Doping is widely used to reduce the phase transition temperature of VO2. Are there any effective doping-elements in reducing the phase transition temperature while without adverse effects on the thermochromic performance of VO2

4) In this review, the authors demonstrate the great potential of VO2 in the application of smart window. So what do the authors think is the biggest advantage of VO2, and what is the biggest obstacle at present?

5) What are the advantages of VO2 smart window compared with the low emissivity glass that is widely used nowadays?

6) One reference has an incorrect format on page 9 of the first sentence. The word “VO2” was not used as subscript in someplace of table 1.

Author Response

Attached is our response.
Best regards.
